# SpatialDM for rapid identification of spatially co-expressed ligand–receptor and revealing cell–cell communication patterns

Zhuoxuan Li[1], Tianjie Wang[2], Pentao Liu ®[1,3] ✉ & Yuanhua Huang ®[1,2,3] ✉

Cell-cell communication is a key aspect of dissecting the complex cellular microenvironment. Existing single-cell and spatial transcriptomics-based methods primarily focus on identifying cell-type pairs for a specific interaction, while less attention has been paid to the prioritisation of interaction features or the identification of interaction spots in the spatial context. Here, we introduce SpatialDM, a statistical model and toolbox leveraging a bivariant Moran's statistic to detect spatially co-expressed ligand and receptor pairs, their local interacting spots (single-spot resolution), and communication patterns. By deriving an analytical null distribution, this method is scalable to millions of spots and shows accurate and robust performance in various simulations. On multiple datasets including melanoma, Ventricular-Subventricular Zone, and intestine, SpatialDM reveals promising communication patterns and identifies differential interactions between conditions, hence enabling the discovery of context-specific cell cooperation and signalling.

Cell-cell communication (CCC) plays essential roles in various biological processes and functional regulations[1,2], for example, immune cooperation in a tumour microenvironment, organ development and stem cell niche maintenance, and wound healing. Protein interaction, as a medium of CCC, has been widely studied in the past decades. Despite the relatively low throughput in proteomics technologies, a large number of ligand-receptor candidates still have been accumulated through broad experimental studies and compiled into databases, e.g., 1396 pairs in CellPhoneDB[3], 1940 pairs in CellChatDB[4] and 380 pairs in ICELLNET[5].

As a more accessible surrogate, the RNAs of ligand and receptor have been shown effective in the quantification of inter-cellular communications[1]. The advancement of single-cell transcriptomics technologies further enables LR interaction (LRI) and CCC in a cell state-specific manner, for example in the maternal–foetal interface[6] and intestinal stem cell niche[7]. Multiple computational methods have soon been developed to identify the interacting cell types and the mediating LR pairs[1,8]. CellPhoneDB is a prominent example that considers multimeric proteins in manually curated LRIs and identifies communicating cell types by comparing the null with permuted cell type labels[3,6]. Another widely used method, CellChat, extends the CCC analysis on multiple aspects, including a mass action model to quantify LR co-expression, expanded LRI candidates with more detailed annotations, and a set of useful plotting utilities[4]. Other methods, including NicheNet[9], PyMINEr[10], iTALK[11], ICELLNET,[5] and SingleCellSignalR[12], have also been introduced in the past two or three years with their unique features on LRI resources and/or testing methods[1]. A recent study further evaluated 16 LRI resources and 7 methods on their impact and consistency in CCC analysis from scRNA-seq data[13], while the direct assessment is generally challenging due to the lack of gold-standard data. Moreover, one major limitation of single-cell-based methods is the lack of spatial coordinates of cells. Therefore, it cannot guarantee physical proximity between the putative interacting cells and may lead to high false-positive rates[8].

In recent years, spatial transcriptomics (ST) technologies have also embraced a few major breakthroughs, on both sequencing and

[1]School of Biomedical Sciences, University of Hong Kong, Hong Kong SAR, China. [2]Department of Statistics and Actuarial Science, University of Hong Kong, Hong Kong SAR, China. [3]Center for Translational Stem Cell Biology, Hong Kong Science and Technology Park, Hong Kong SAR, China. ✉e-mail: pliu88@hku.hk; yuanhua@hku.hk

imaging-based platforms[14]; therefore, ST is increasingly used to double-check the physical proximity of the LRI identified in single-cell data. Meanwhile, a few ST-based methods have been developed to identify CCC and LRIs directly from ST data[15,16]. Giotto is a toolbox for multifaceted analyses of ST data, including detecting cell-type pairs that have increased interactions of proximal cells than those at random locations[17]. scHOT[18] and SpatialCorr[19] respectively introduced weighted Spearman's or Pearson's correlation to test gene correlations for each spatial pixel and select gene pairs (or sets) with differential correlation across space. SVCA is a Gaussian process-like method that defines a universal cell-cell interaction covariance over spatially smoothed cell embeddings and consequently identifies genes with a high proportion of variance explained by this interaction term[20]. SpaOTsc leverages an optimal transport method to quantify the likelihood of interaction between any two cells, with spatial distance as one cost component[21]. SpaTalk is another recently proposed toolbox to analyse spatial LRI and CCC by testing if a certain cell-type pair is enriched in those co-expressed spots[22]. Although these methods brought promise to directly analyse CCC in a spatial context, most of them focus on identifying interacting cell types for all LRIs instead of detecting the interacting LR pairs first, hence may over-interpret less informative LRIs. Additionally, most of these strategies may not be sensitive enough to identify regional CCC, as they aim to detect cell types with enriched interactions as a whole (Fig. 1a). Moreover, the conventional permutation test is not scalable and may slow down the computational analysis, particularly considering the fast advances in spatial resolution and cell numbers.

Here, to address these limitations, we introduce SpatialDM (Spatial Direct Messaging, or Spatial co-expressed ligand and receptor Detected by Moran's bivariant extension), a statistical model and toolbox that uses a bivariate Moran's statistic to identify the spatial co-expression (i.e., spatial association) between a pair of ligand and receptor. Critically, we introduced an analytical derivation of the null distribution, making it highly scalable to analyse millions of cells. This method also contains effective strategies to identify interacting local spots and the patterns shared by multiple LRIs or pathways. We evaluated the accuracy of SpatialDM with various simulations and demonstrated its broad applicability in detecting LRIs and differential interactions between conditions in melanoma and intestinal datasets by high-throughput sequencing and in a mouse SVZ dataset by Fluorescent In Situ Hybridization (FISH, Supplementary Fig. 1).

## Results

### Overview of SpatialDM method

Identification of the communicating cells and the interacting LR pairs are the two major orthogonal tasks in dissecting CCC in scRNA-seq and ST data. Most existing methods mainly aim to address the former challenge (at cell-cluster or cell-type resolution) but omit the latter task of feature selection simply by relying on a curated database. However, we argue that identifying the dataset-specific interacting LR pairs is a crucial step for ensuring quality analysis and reliable interpretation of the putative CCC.

Therefore, the primary aim and the first step of SpatialDM is to detect LR pairs that have significant spatial co-expression (i.e. ligand and receptor transcripts are expressed within a reasonable geographical distance) in ST data. The candidate LR pairs are generally from a comprehensively curated database, e.g. CellChatDB by default. Figure 1a shows an example that the LR pair B has spatial co-expression and can be detected by SpatialDM, while pair A does not though its cluster-level enrichment may lead to false positives in existing approaches. Generally, this problem of spatial association between two variables can be formulated by a regression model, either via fixed effects, e.g., SDM and SDEM[23] or random effects, e.g., SVCA[20]. Here, we introduce a bi-variate Moran's $R$ as a test statistic (Fig. 1a; "Methods" section), which can well account for the spatial association, i.e., the

spatial co-expression of ligand and receptor here. This method is an extension of the well-known Moran's $I$ in uni-variate auto-correlation analysis[24] to a bivariate setting initially by Wartenberg[25] and is still widely used in the broad field of spatial analysis[26,27]. The computational convenience and effectiveness make it an appealing method for LRI in ST data (see evaluation below).

As a computational toolbox, SpatialDM has major functions for both global and local analyses (Fig. 1b). First, by leveraging this bivariate $R$, we introduce a hypothesis testing to reject the null that the ligand and receptor are spatially independent, hence allowing us to select the spatially co-expressed LR pairs. Second, we further adapted local Moran's $I$ to their bivariate format to detect local hits for each significant LR pair (Methods). Based on the local interaction hits for each LR pair, SpatialDM allows grouping these significant LR pairs into a few distinct communication patterns, e.g., by the automatic expression histology model introduced in SpatialDE[28]. Third, to interpret the local communication patterns, it also provides an enrichment test and visualisation of putative pathways for each local pattern. Last, as a unique feature, SpatialDM further supports detecting LR pairs that have differential interaction density between conditions or along a continuous covariate, which is highly demanded for biological discovery in both developmental and disease contexts.

### Accurate and efficient z-score test

In order to obtain the null distribution in this hypothesis testing problem, a generic method is permutation as used by most CCC methods, where the test statistic $R$ will be calculated by random shuffling of binding partners for each pair, e.g., 1000 times. On the other hand, when the number of spatial spots is large, the permutation test often becomes a computational bottleneck for the analysis. Therefore, we derived the first and second moments of the null distribution to analytically obtain a z-score and its according $p$-value for the observed $R$ (see Supp. Note 1). Strikingly, the z-score-based $p$-value has high correlations with the permutation-based $p$-value in datasets with different sizes (Fig. 1c, d; Spearman's $R > 0.9$, local statistics correlation: Supplementary Fig. 2d, e). Given the computational convenience, SpatialDM (the permutation mode, 1 CPU) ranks as the fastest method among all permutation-based methods, finishing testing 1000 LR pairs within 1.5 min for a 10,000-spot dataset (even though all other methods using 50 CPUs except SpaTalk and SpatialCorr). Importantly, the z-score-based strategy further introduces over 100x speedups, therefore is exclusively scalable to a million spots within 12 minutes (even with a single CPU). Therefore, this innovation of analytical null distribution can be highly valuable for the analysis of ST data with increasingly large sizes.

To examine the accuracy of SpatialDM in detecting spatially correlated ligand-receptor pairs, we first generated multiple sets of simulated ST data by adapting a recent method SVCA[20]. In short, SVCA is a principled Gaussian process model that decomposes the variance of a certain gene (a ligand here) into cell states, spatial proximity, spatially weighted receptor (i.e., the ligand-receptor spatial interaction), and residual noise (see "Methods" section). Here, based on a seed dataset with 293 spots and 1180 LR pairs, we first generated a negative set with 0% variance explained by ligand-receptor spatial interaction. When applying SpatialDM to this negative data set (under the null), we found that the $p$-values of both permutation and z-score are well calibrated to a uniform distribution (Fig. 1f), despite the data being generated by a different model. In contrast, CellChat with 2 different parameter settings[4], Giotto[17], and SpaTalk[22] failed to control false positives, as they work for different purposes.

To further evaluate the power of SpatialDM and its overall performance, we generated a positive set with 25% variance explained by the spatial correlation ("Methods" section) and applied SpatialDM to the pool of positive and negative sets. With the default cutoff of $p$-value < 0.05, SpatialDM achieves a power of 74.5% and controls a false

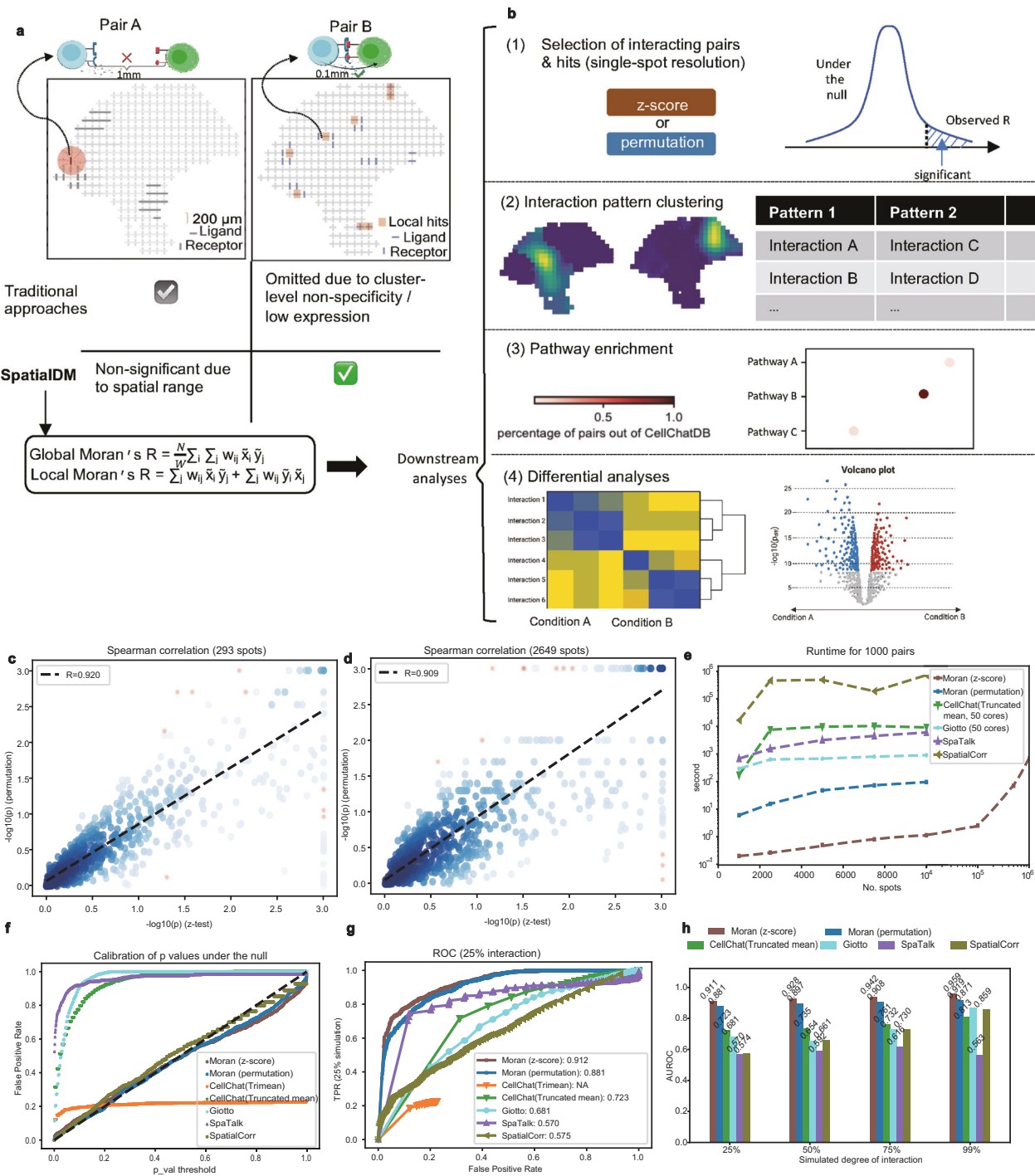

**Fig. 1 | SpatialDM provides a LRI toolkit with high specificity and sensitivity.**
**a** Illustration of SpatialDM method. Top: two examples of ligand-receptor pairs in ST where pair A are barely likely to interact due to distant location while pair B can. Different from traditional approaches focusing on cell type enrichment, SpatialDM aims to detect spatially co-expressed LR pairs by a bivariate Moran's *R*. **b** The four major utility functions in the SpatialDM toolbox for CCC analysis, including (1) selecting LR pairs with global significance, local interaction hits and (2) pattern classification, (3) pathway enrichment analysis and (4) detecting differential interacting pairs between conditions. **c**, **d** Consistency between SpatialDM's permutation and *z*-score modes in a melanoma slice (293 spots; **c**) and an intestine slice (2649 spots; **d**), respectively. Spearman correlation coefficients (two-sided) were specified on the top left. Fitted linear regression is in a black dashed line. Source

data are provided as a Source Data file. **e** Running time comparison of SpatialDM with CellChat, Giotto, SpaTalk and SpatialCorr, where only SpatialDM (*z*-score mode) is scalable to 1 million spots within 12 min with one CPU core. Source data are provided as a Source Data file. **f**, **h** Assessment by simulated datasets. Source data are provided as a Source Data file. All methods use one-sided tests. **f** False Positive Rate (FPR) comparison when no interaction is simulated; SpatialDM and SpatialCorr calibrate with the null distribution along with the diagonal line.
**g** Receiver Operating Characteristic (ROC) under the 25% degree of interaction simulation scenario. Area Under ROC (AUROC) for each method is labelled in the legend. **h** Comparison of AUROC under four different degrees of interactions (25%, 50%, 75%, 99% interaction respectively).

positive rate of 8.2% with the *z*-score approach. By varying the *p*-value, it returns an AUROC of 0.912 (*z*-score mode; permutation AUROC=0.881), demonstrating its unique advantage in detecting spatially correlated LRIs under the simulation scenario (other methods' AUROC: 0.570 to 0.723; Fig. 1g). Similar results were also observed when generating positive samples with higher levels of variance explained by spatial interaction from 50%, 75% to 99%, where the AUROC increases accordingly up to 0.959 (Fig. 1h and Supplementary Fig. 2a–c). Note, all methods in comparison may not be favoured by the simulation setup with the objective to capture spatially co-varying ligand-receptor interactions. As spatial data is generally sparse and CellChat's Trimean mode might be too stringent to generate a high power (Fig. 1g and Supplementary Fig. 2a–c), we excluded it for further comparison and only kept CellChat's Truncated-mean mode.

## Detecting spatial LRI in melanoma

Next, we applied our SpatialDM to the aforementioned seed data, a melanoma sample probed by ST platform (200 μm centre-to-centre distance), covering over 7 cell types from 293 spots[29]. Given the small sample size, we employed SpatialDM's permutation approach. When applying to the 1180 LR pairs from CellChatDB, SpatialDM detects 103 spatially co-expressed pairs (FDR < 0.1; Fig. 2a and Supplementary Dataset 1). In contrast, other methods generated 340–874 significant pairs except SpatialCorr (75 pairs), raising the possibility of false positives (Supplementary Dataset 1). Indeed, all other methods suffer from high false positives when testing on a manually generated negative set by shuffling the ligand–receptor database to create a list

of 663 non-documented ligand–receptor pairs (e.g., 285 pairs by Giotto as the best counterpart; Supplementary Fig. 3a and Supplementary Dataset 2). However, SpatialDM and SpatialCorr have good false-positive controls here (90 and 80 pairs, respectively; permutation *p*-value < 0.05), which is consistent with the simulation (Supplementary Figure 3a and Fig. 1f). A similar pattern is also observed on two expected irrelevant LR pairs (*FGF2_FZD8* and *PHF5A_EDEM3*; Fig. 2a).

Interestingly, many known melanoma-related genes like *VEGF*, *SPP1*, and *CSF1* have been included in the 103 LR pairs selected by SpatialDM. Further, we applied SpatialDM to identify local hits of interaction by local Moran's *R* (*p* < 0.1). Given the general low depth in spatial transcriptomics, the method proves sensitive enough by detecting pairs as sparse as 2 interaction spots, and also powerful by detecting as many as 72 spots. The 103 selected pairs were subjected to automatic expression histology from SpatialDE, which resulted in 3 coarse patterns (Fig 2b and Supplementary Dataset 3). We observed that Pattern 0 corresponds to the lymphoid region, Pattern 1 simulates the melanoma region, and Pattern 2 maps to the cancer-associated fibroblast (CAF) region, referenced to Thrane et al.[29] and the predicted cell types from scRNA-seq by RCTD[30] (Fig. 2b). Indeed, we found that the local interaction scores are good predictors of the cell types (Pearson's *R* = 0.928; linear regression; Supplementary Fig. 3b).

We then identified pathways enriched in each pattern, and found that the melanoma region (i.e. pattern 1) shows signatures of angiogenesis and tumour progression (Supplementary Figs. 3c and 4 and Supplementary Dataset 3). Immunity-related pathways (including CCL and CD23) were enriched in the lymphoid region (i.e. pattern 0, Fig 2c

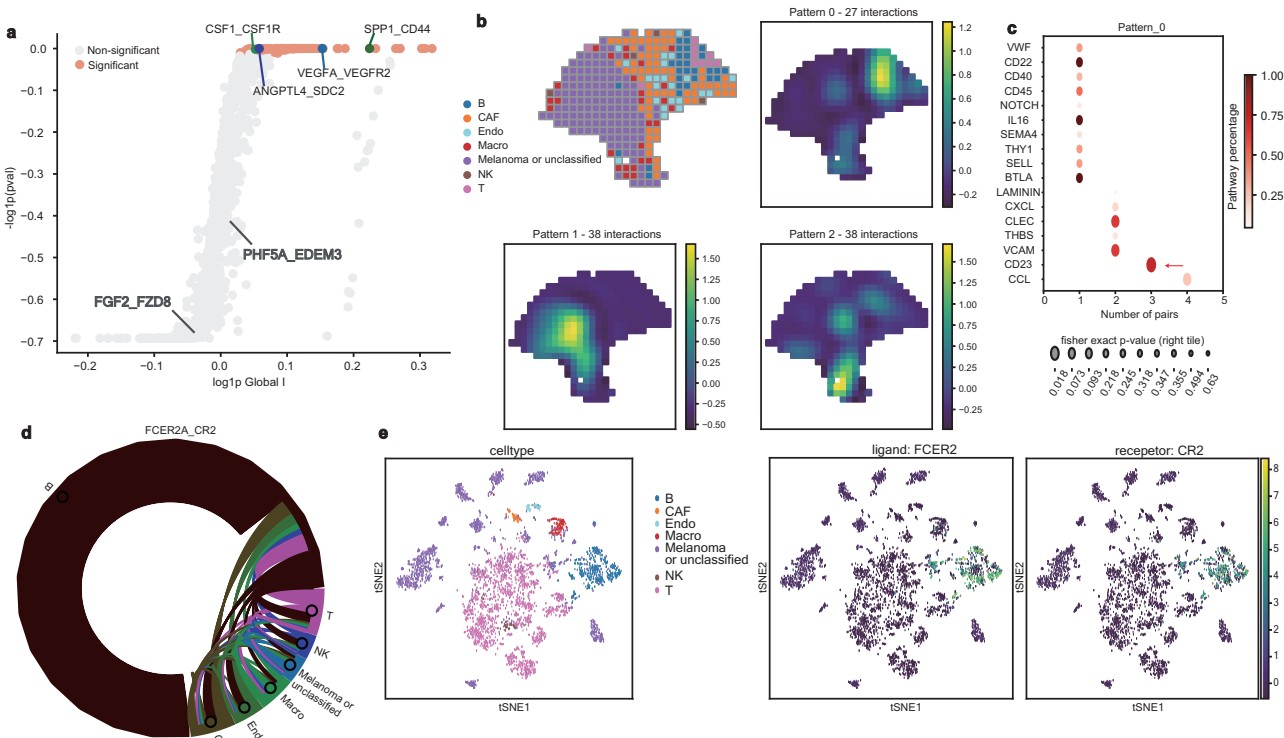

**Fig. 2 | SpatialDM detects spatially co-expressed LRs in melanoma data and identifies CCC patterns. a** Scatterplot of Global Moran's *R* and permutation *p*-value (one-sided). *X*-axis is log(Global Moran's R+1); *y*-axis is -log(*p* value+1). Significant pairs (FDR < 0.1) are highlighted in orange. Four well-known positive samples (coloured) and two negative samples (in grey) are highlighted. **b** Upper left: spatial plot coloured by cell types that have the largest proportion estimated by RCTD. Source data are provided as a Source Data file. Right: Clustering of 103 significant pairs into three spatial patterns by SpatialDE. A representative plot is shown for each spatial pattern. Each plot is coloured by the posterior mean of local statistics in each spatial pattern. **c** Dot plots of enriched pathways in pattern 0 from panel **b**. *Y*-

axis: the name of the enriched pathway; x-axis: the number of significant LR pairs in that pathway for each pattern; dot colour: the percentage of significant pairs for that pathway from CellChatDB; dot size: significance of enrichment from One-sided Fisher's Exact Test ("Methods" section). **d** Chord diagram summarising cell types interacting for FCER2A_CR2 interaction. Cell types are distinguished by node colours, and edge colours indicate the sender cell types. **e** tSNE plots of matched melanoma scRNA data, coloured by the original cell types and expression level of ligand FCER2 and receptor CR2, one pair of CD23 ligand-receptor interaction pathway as highlighted in **c**.

and Supplementary Fig. 4), concordant with histology annotations provided by the authors and RCTD annotated results (Fig. 2b, c and Supplementary Dataset 3). CD23, a less-discussed pathway in melanoma showed high relevance in pattern 0 (Fig. 2c), which led us to examine the result in an annotated melanoma scRNA-seq dataset with greater sequencing depth and resolution. *CD23* (a.k.a, *FCER2*) could bind with *CR2* or integrin complexes to trigger immunologic responses[31,32]. Consistent with the identified region (pattern 0), it was mainly found in B cells (Fig. 2d). In another melanoma scRNA-seq data we examined[33], *FCER2* and its receptors were also enriched in the B cells, which is 20-fold higher than any other cluster, validating the discoveries from spatial transcriptomic analyses (Fig. 2d, e and Supplementary Fig. 3d). Interestingly, by examining the 65 genes differentially expressed in the CD23 hot spots, we found that they are highly enriched in immune cell activation pathways, supporting anti-tumour functions, instead of a pro-tumour role (Supplementary Fig. 3e and Supplementary Dataset 4). Taken together, these identified LRI and their regional patterns may contribute to further signalling investigation and potential treatment targets.

### Identifying consistent cell–cell communications in multiple intestine samples

Human intestines originate from all three germ layers, involving a variety of developmental cues at different post-conceptual weeks (PCW), and sophisticated self-renewing mechanisms of the crypt-villus structure throughout adult life. With time-stamped single-cell and spatial transcriptomic datasets from 12 post-conceptual weeks (12 PCW: 3 colon replicates from 2 donors, A3, A8 and A9, 2 TI replicates from one donor, A6, A7) or 19 PCW foetus sample (1 slice, A4) to adult samples (2 replicates from 1 donor, A1 and A2, with IBD or cancer), Corbett, et al. have identified several ligand-receptor interactions through customised analyses (100 µm spot-spot distance, Supplementary Dataset 5)[7]. Briefly, Corbett, et al. screened through a database of over 2,000 LR pairs, giving each ligand and receptor specificity scores and expression scores across each of the 101 scRNA clusters; then, the putative list of LR interactions with high specificity and expression in a cluster-cluster combination was validated in spatial transcriptome regarding LR spatial co-localisation. As a result, Corbett, et al. have identified *CEACAM1_CEACAM5* toward the crypt top in adult samples, *IL7_IL7R_IL2RG*, *CCL21_CCR7* and *CCL19_CCR7* between Lymphoid Tissue Inducer (LTi) and S4, *ANGPT2* in foetal vasculature, and many others[7]. Considering the large sample size, we leveraged the *z*-score approach in SpatialDM to re-analyse all samples in this dataset, and identified majority of these reported interacting pairs (326 out of 414; Supplementary Dataset 6 and Supplementary Fig. 5a). More interestingly, 220 additional LR pairs are uniquely identified by SpatialDM, suggesting its potentially enhanced sensitivity in detecting sparsely expressed LR pairs.

Thanks to the multi-sample setting, we first used this dataset to assess the reproducibility of SpatialDM in both detecting spatially co-expressed LR pairs and their communicating regions. When comparing the global Moran's *R*, we observed high correlations between slices from the same sample versus low correlations among slices from different samples (Fig. 3a). Similarly, whole-interactome clustering revealed the dendrogram relationships that are close to the sample kinship (e.g. A8 and A9 from one 12 PCW sample is close to another 12 PCW sample A3 but far from the adult samples A1 and A2; Supplementary Fig. 5b, c).

Next, we assessed whether local hits discovered by SpatialDM are consistent in technical or even biological replicates. The cell type weights of local selected spots are highly correlated between technical replicates (e.g. median Pearson's $R = 0.975$ for A1 vs. A2 and $R = 0.862$ for A8 vs. A9, Supplementary Fig. 5d–f), moderately correlated between biological replicates (e.g. A3 vs. A9), but poorly correlated in distinct samples (e.g. A3 vs. A7, Supplementary Figure 5g). Given the

sensitivity of SpatialDM, the consistency in local pattern detection is observed for both ubiquitously interacting pairs and sparse ones, from which we illustrate two concrete examples here. *FN1_CD44* interacts more ubiquitously in adult and foetus colons (Supplementary Figs. 5d and 6 and Supplementary Dataset 6), probably due to its versatile role during intestine development[34]. The interaction of *PLG_F2RL1* is sparsely found in all foetal slices, and with consistent cell-type enrichment in enterocytes (Fig. 3b and Supplementary Fig. 6).

### EGF pathway interactions are enriched in adult crypt top colonocytes

Seeing the consistency of SpatialDM between technical replicates, we then zoomed into sample A1 to reveal the interaction patterns in adult colons with IBD or cancer. Through similar procedures as in melanoma analysis, the 362 significant pairs (*z*-score FDR < 0.1, hits in at least 10 spots) were classified into 4 patterns (Supplementary Dataset 7). Pattern 1 is mostly enriched in immune cells, pattern 2 in crypt top colonocytes, and pattern 0, 3 in myofibroblast (Fig. 3c, d and Supplementary Fig. 7a). Such cell-type enrichment patterns are consistent with pathway enrichment. For example, interactions under MHC-II and ICAM pathways show high relevance in pattern 1, which showed enrichment in immune cells, suggesting an inflammatory microenvironment in the adult colon (Supplementary Fig. 7b and Supplementary Dataset 7). The EGF pathway comprises diverse ligands (including *EGF*, *TGFA*, *AREG*, *EREG*, and *HBEGF*) and receptors (including *EGFR*, *ERBB2*, *ERBB3, and ERBB4*), exerting distinct or redundant functions[35]. In the adult sample we analysed, most EGF interactions were detected and enriched in pattern 2 (Fig. 3e, f, Supplementary Fig. 7C and Supplementary Dataset 7).

The EGF signalling plays important roles primarily in intestinal epithelial cell proliferation and self-renewal, and has a complex interplay with other pathways[35]. Nászai, et al. have revealed that RAL GTPases, encoded by RALA and RALB, are necessary and sufficient to activate EGFR signalling and further MAPK signalling in the intestine[36]. Interestingly, we indeed found that the upstream RALA and RALB expression and downstream MAPK expression have great overlap with the local Moran selected spots (Fig. 3g and Supplementary Fig. 7d). It highlights the potential to detect interplays with upstream or downstream signalling of LRI captured by SpatialDM.

### SpaitalDM identifies differential interactions between foetus and adults

Besides the sample-independent analysis, SpatialDM allows differential analysis of detailed interactive pairs between conditions or along with a continuous covariate, accounting for multiple replicates. Briefly, a (generalised) linear model is introduced to test if a certain covariate affects the interaction density (indicated by the *z*-score or permutation numbers; see "Methods" section). Here, we showcase the differential analyses among adult vs. foetal colon samples based on the *z*-score inputs (Fig. 4a, b and Supplementary Dataset 8), where 146 pairs of LR interactions are up-regulated in adult samples while 97 pairs in the foetus (FDR < 0.1; likelihood ratio test, Fig. 4b).

By pathway enrichment analysis (Fig. 4c), we first noticed the adult-specific pairs enriched with chemokine and cytokine responses (e.g. ICAM, CCL and CXCL) as well as inflammatory and immune signatures (e.g. MHC-II, COMPLEMENT, BMP and MIF), which is consistent with insights from previous comparative RNAseq analysis[37]. It was known that inflammation in the foetus can be associated with preterm parturition[38], Fetal Inflammatory Response Syndrome (FIRS)[39], impaired neurological outcomes[40], and other defects. In our analysis, some pathways like COMPLEMENT are generally exclusive in adults, while other interactions like TGFB and CCL can be possibly established early in the foetus stage. For example, *TGFB3_TGFBR1_TGFBR2* was identified across each time point (Supplementary Dataset 8). TGFBs are potent immunosuppressive

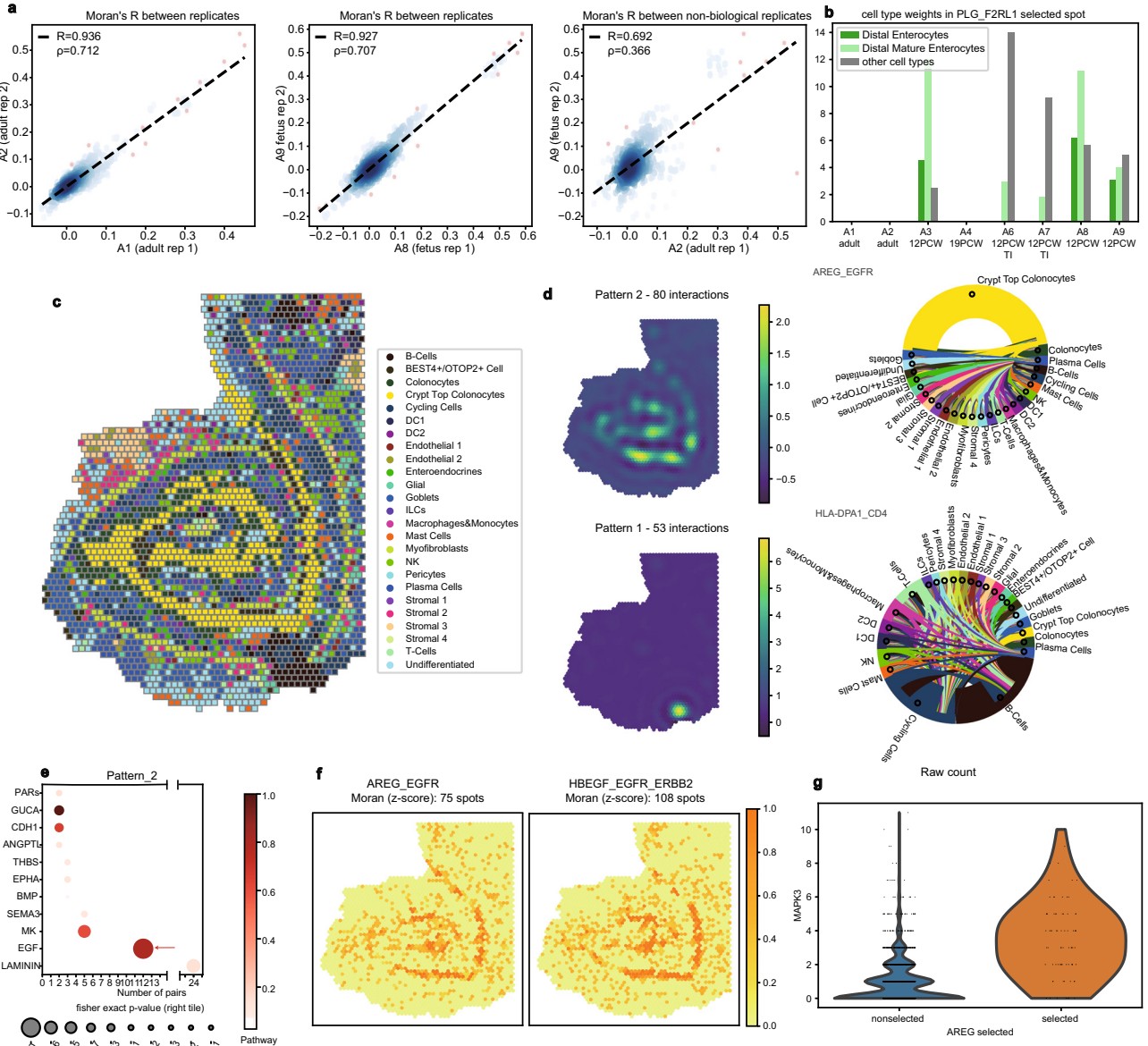

**Fig. 3 | Multiple intestinal samples for technical validation and CCC pattern discovery. a** Cross-replicate correlations of Global Moran's *R*. Pearson coefficients R and Spearman coefficients ρ were specified in the top left. Source data are provided as a Source Data file. (From left to right) Correlations between two adult replicates (A1 and A2), between two foetuses, replicates (A8 and A9), and between an adult and a foetus sample. **b** Scaled cell type weights of *PLG_F2RL1* interacting spots (local z-score *p* < 0.1, one-sided). *PLG_F2RL1* was not identified in adult samples (A1, A2) and 19 PCW colons (A4). Other cell types than enterocytes have been grouped together. **c** Spatial plot of A1 sample coloured by cell type annotated from the original study. **d** Summary of pattern 2 and 1 in A1 from SpatialDE. Left: Spatial plots of the pattern. Right: chord diagram for cell type compositions for the selected pairs. Node size indicates the summation of the decomposite cell type weights over selected sender or receiver spots. The edge colour indicates the sender cell type. **e** Pathway enrichment result for pattern 2 in A1 (one-sided). **f** Selected local spots of two example pairs under EGF pathways. Colour denotes 1−*p*. Dot size denotes Fisher's Test significance. The numbers of significant spots were specified (*p* < 0.1, one-sided). **g** Comparison of MAPK3 expression in AREG_EGFR local selected spots (z-score *p* < 0.1, one-sided) vs. non-selected spots.

cytokines, which drive the functional development of lymphocytes, therefore reinforcing the gut barrier. Such interactions may have critical roles during early intestine development at the foetus stage.

We also observed that the foetus-enriched pathways are associated with neural processes (e.g. NRXN, GDNF, PTN), new blood vessel formation (e.g. SEMA, VEGF), and growth (e.g. GDF, MK; Fig. 4d). Such observations of early establishment prior to 12 PCW were consistent with Corbett, et al.[7]. Overall, we provide evidence that the diseased adult intestine has a more pro-inflammatory environment, while the foetal intestine has more development-related signatures.

Beyond pathway-level comparison, SpatialDM allows differential analysis on a certain ligand-receptor pair (Supplementary Dataset 8).

While traditional pathway enrichment may have ignored BMP pathway enrichment in adults, SpatialDM refines the adult-specific interactions to *BMP2* and its receptors (BMPR1A/B and ACVR2A, Supplementary Dataset 6 and 8). In fact, with the function of promoting apoptosis and inhibiting proliferation, *BMP2* was previously revealed by RT-PCR and immunoblotting to be expressed by, and act on mature colon epithelial cells[41]. There have also been multiple reports of epithelium-immune orchestration in the adult intestine. We have identified *NRG4_ERBB2* among various cell types in adult-specific interactions (FDR < 0.0001, Fig. 4b), but not in foetus samples. Interestingly, NRG4 was found in human breast milk, and its oral supplementation can protect against inflammation in the intestine[42]. Our analysis

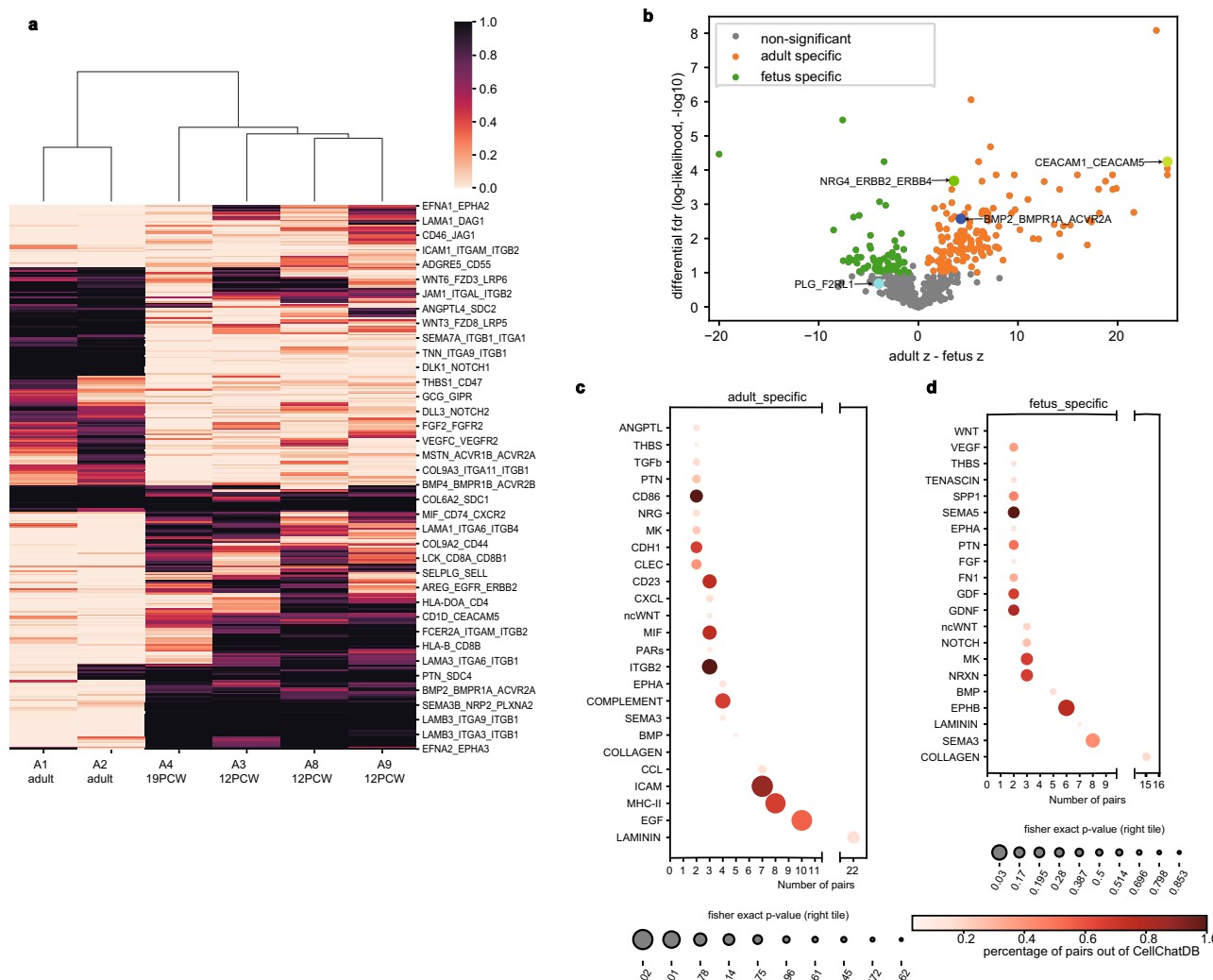

**Fig. 4 | SpatialDM identifies differential LR interaction between foetus and adult intestines. a** Heatmap coloured by global z-score *p*-values of adult-foetus differential pairs (likelihood ratio test FDR < 0.1, two-sided), including 135 adult-specific (low *p*-values in A1 and A2), and 98 foetus-specific. **b** Volcano plot highlighting adult-specific and foetus-specific pairs, in orange and green, respectively (FDR < 0.1). z difference (*x*-axis) cutoffs were taken at 30% and 70% quantiles, respectively. Source data are provided as a Source Data file. **c, d** Pathway enrichment results for adult-specific and foetus-specific pairs. Dot sizes indicate enrichment significance from one-sided Fisher's Exact Test results. Only pathways with 2 or more selected pairs were visualised.

consolidated that certain anti-inflammation mechanisms may only be established after early conceptual weeks, likely at infant breastfeeding stages as reported.

In addition to adult-only pairs, our differential analysis allows the detection of LR pairs with a subtle but significant change in communication density between adult and foetus (Fig. 4b). *CEACAM1_CEACAM5* is an example that was also demonstrated in adult samples by the authors. Although we have identified *CEACAM1_CEACAM5* in A3 with a moderate signal (FDR = 0.006) in addition to two adult slices, *CEACAM1_CEACAM5* was considered adult-specific in the differential analysis (Fig. 4b, differential *p* < 0.0001, A1 *R* = 0.433, A2 *R* = 0.577). In fact, *CEACAM1_CEACAM5* is only sparsely expressed in A3 (*R* = 0.034), with few positive significant interaction spots (Supplementary Figure 7e). Both molecules were recognised to be highly present in human colon epithelia and related to inflammation and tumorigenesis[43]. Defects in CEACAM signalling in intestinal epithelial cells are associated with Inflammatory Bowel Disease (IBD), and even Colitis-Associated Cancer (CAC)[43,44]. As we revealed the interplay of various cell types including colonocytes and cycling cells in *CEACAM1_CEA-CAM5* interaction in IBD or colorectal cancer patients, it might highlight targeting these cells to reverse the adverse conditions.

Overall, SpatialDM has not only validated a number of interactions discussed by the original report, but also uncovered multiple insights into the inter-compartment orchestrations in the human intestine, especially by allowing differential analyses among multiple replicates. Therefore, SpatialDM enables the generation of new hypotheses for further experimental studies to discover more underlying mechanisms of intestinal disorders which are currently poorly understood.

## Discussion

To tackle unaddressed questions in spatial transcriptome as to what ligand-receptor interact and where they take place, we introduce SpatialDM, a statistical model in the form of bivariate Moran's method. This method uniquely aims to effectively detect the spatially co-expressed ligand-receptor at single-spot resolution as the primary task, ensuring the high-quality discovery of communication patterns. Critically, we also derived an analytical form of the null distribution, therefore SpatialDM does not need to rely on the time-consuming permutation test, and is scalable to millions of spots.

Following the significant LR pairs, SpatialDM further identifies the local communicating spots and their regional patterns, facilitating various downstream explorations. Notably, the concise framework

also allows differential analyses under multi-sample settings with the likelihood-ratio test of global *z*-scores. This facilitates spatial-temporal analyses of cell-cell interactions in a time-series design or along with a pseudo-time trajectory. Such differential analyses are not only helpful in identifying disease mechanisms and potential treatment targets but also enable the detection of subtle changes during development on an interacting pair level instead of the pathway level.

Similar to most CCC methods, SpatialDM also takes a curated LR database as input. As SpatialDM is capable of detecting dataset-specific LR pairs, we generally recommend feeding a more comprehensive database, e.g., CellChatDB by default, while one can input a customised candidate list. Of note, all analyses of ST data here are only on the mRNA level, while other factors, e.g., alternative splicing, translation machinery, and post-translational modifications can further determine whether the interactions actually happen outside the cell. While ST datasets have been examined in this paper given their prevalence, the same framework could, in principle, be directly applied to high-throughput spatial proteomic datasets to facilitate more direct interpretations, particularly considering the rapid development of spatial proteomics or multi-omics technologies, e.g., Deep Visual Proteomics (DVP) and DBiT-seq[45,46].

Another open challenge is to identify the downstream targets of LR interactions, which can largely enhance the interpretation of the signalling pathway of a certain CCC. Though we showed one case that the literature-reported downstream targets are well supported here, a comprehensively curated database with high quality will be largely appreciated to perform a systematic investigation; the scMLnet database might be an option[47]. Additionally, more sophisticated methods are desired in addressing this challenge.

Furthermore, there are also technical elements in the SpatialDM framework worth further exploration. First, we only used the RBF kernel for defining the spatial similarity matrix, while other kernels may be applicable too, e.g., the Cauchy kernel or a mixture of multiple kernels. Second, although we have demonstrated the effectiveness of detecting local interaction hits, the local Moran's *R* value is not normalised to a fixed bound, e.g., (−1, 1), but it is refined to a reasonable range after standardising the expression matrices (i.e., −10 to 10), and we have further clipped the extreme values out this range. Therefore, the standardisation of local *R* values makes all ligand-receptor pairs comparable, both within and across samples. On the other hand, this standardisation has a minor sacrifice by losing the information on the local communication density of each pair (namely some pairs may have higher expression than others). In certain scenarios where the original expression level of the ligand-receptor pair is highly informative, one can turn off the standardisation when interpreting the local *R* values. Nonetheless, the hypothesis testing and its *p*-value are robust to the setting with or without standardisation. Another relevant challenge is to simulate realistic data that both holds the global structure and the interaction patterns of each spot; we anticipate more sophisticated simulators will be proposed to enhance local hits detection in near future. Third, given a small number of replicates, the detection of differential communicating LR pairs between conditions is generally challenging, hence a Bayesian treatment for jointly analysing all pairs may mitigate this issue to a certain degree. Last, another potential limitation is that the pair-independent analysis in SpatialDM may oversimplify communication events due to potential pleiotropy between ligands where multiple ligands interact with the same receptor.

To conclude, the method presented here resolved the selection of the spatially communicating LR pairs in ST data, allowing for effective CCC pattern discovery in a local region and identification of condition-specific communications. With the rapid development of spatial omics technologies, SpatialDM opens up an efficient and reliable way to dissect cell cooperation in a micro-environment.

## Methods

### Global Moran's *R* for spatial co-expression

In order to analyse reliable cell-cell communication in ST data, SpatialDM aims to identify ligand-receptor with significant spatial co-expression, from a comprehensive candidate list. By default, we use LR lists from CellChatDB v.1.1.3 (mouse: 2022 pairs, human: 1940 pairs, zebrafish: 2774 pairs) as input[4], while users can use any customised list.

Here, for detecting the spatial co-expression, we extended the widely used Moran's *I* from a univariate to a bivariate setting. This is an extension which is closely related to the earlier use in geography proposed by Wartenberg[25]. In order to distinguish the spatial auto-correlation in a univariate setting, we call this bivariate statistic Moran's *R*, as follows

$$Global\ Moran's R = \frac{\sum_i \sum_j w_{ij}(x_i - \bar{x})(y_j - \bar{y})}{\sqrt{\sum_i (x_i - \bar{x})^2}\sqrt{\sum_i (y_i - \bar{y})^2}},\qquad (1)$$

where $x_i$ and $y_j$ denotes normalised and log-transformed ligand and receptor expression at spot *i* and *j*, respectively. Spatial weight matrix computation is based on Radial Basis Function (RBF) kernel with an element-wise normalisation,

$$w_{ij}^{(0)} = \exp\left\{-\frac{d_{ij}^2}{2l^2}\right\}; w_{ij} = \frac{n}{W} w_{ij}^{(0)},\qquad (2)$$

where $d_{ij}$ is the geographical distance between spot *i* and *j* (i.e., Euclidean distance on spatial coordinates), *W* is the sum of $w_{ij}^{(0)}$, and *n* is the number of spots. Optionally, if assuming single-cell resolution, the diagonal of the weight matrix can be made 0 to reduce the influence by auto-correlations, namely $w_{ii} = 0$ for any *i*. For the analysis in this work, the SVZ dataset is supposed to be of the single-cell resolution, while melanoma and intestine datasets are not.

In addition to the scale factor *l* in the RBF kernel, alternative options through either cut-off (co) or the number of nearest neighbours (n_neighbors) can be customised to restrain secreted signalling within certain spots' diameter distance. In the melanoma data (200 μm centre-to-centre distance) analysis, we assigned *l* = 1.2, co = 0.2; In the intestine data (100 μm centre-to-centre distance) analysis, we assigned *l* = 75, co = 0.2 (according to larger coordinate scale); In the SVZ data (single-cell resolution), we assigned *l* = 130, co = 0.001. Such settings are based on the assumption that secreted signalling can occur in 100–200 μm (i.e. 1199 pairs secreted signalling in CellChatDB-human), although signalling of longer distances may not be tracked (e.g. hormone). For short-distance signalling (i.e. 421 ECM-receptor pairs or 319 cell–cell contact pairs in CellChatDB-human), another weight matrix is implemented (nearest_neighbors) which limits the interaction to the most adjacent cells (default 6 cells).

For ligands or receptors composed of multiple subunits, we computed the algebraic means as inputs for SpatialDM, i.e.

$$x_i = \frac{\sum_{s=1}^{S_L} x_i^{(s)}}{S_L}; y_j = \frac{\sum_{s=1}^{S_R} y_j^{(s)}}{S_R},\qquad (3)$$

where *s* is the $s_{th}$ subunit for ligand $x_i$ (with $S_L$ subunits) or receptor $y_j$ (with $S_R$ subunits). Users can also opt for geometric means for more stringent selection results.

### Hypothesis testing with global Moran's *R*

In order to perform the hypothesis testing, the distribution of *R* statistic under the null (i.e., ligand and receptor are spatially independent). Two methods can be adopted to approximate the null distribution and calculate the *p* value: (1) Permutation method by shuffling $w_{ij}$ for multiple times (e.g. 1000), and then calculate the *p* value as the proportion of the permutation *R* values that are as large as

the observed value; (2) Analytical method by approximating the null distribution with a normal distribution by deriving its first and second moments (see Supp. Note 1), then a corresponding z-score can be calculated, as follows:

$$z = \frac{R - 0}{\sqrt{\mathrm{Var}(R)}}, \qquad (4)$$

where the final form of the variance can be written as:

$$\mathrm{Var}(R) = \frac{n^2 \sum_{i=1}^{n} \sum_{j=1}^{n} w_{ij} w_{ji} - 2n(\sum_{i=1}^{n}(\sum_{j=1}^{n} w_{ij} \sum_{j=1}^{n} w_{ji}) + (\sum_{i=1}^{n} \sum_{j=1}^{n} w_{ij})^2}{n^2(n-1)^2} \qquad (5)$$

Then the $p$-value can be obtained by the survival function in a standard normal distribution from the z-score.

## Significant interaction spots

Similar to the global $R$, we also introduce local $R$ in a bivariate setting as a testing statistic to indicate the local interacting spots for each ligand-receptor pair. The local Moran's $R_i$ for spot $i$ is composed of sender statistics and receiver statistics and defined as follows,

$$Local\ Moran's R_i = R_{i,sender} + R_{i,receiver} = x_i' \sum_{j=1}^{n} w_{ij} y_j' + y_i' \sum_{j=1}^{n} w_{ij} x_j', \quad (6)$$

where $x'$ and $y'$ denotes gene-wise standardised (i.e. $x_i' = \frac{x_i - \bar{x}}{\sigma_x}$, $y_i' = \frac{y_i - \bar{y}}{\sigma_y}$; same as `scanpy.pp.scale`) ligand and receptor expression, respectively.

Similar to the Global counterpart, we applied both permutation and z-score approaches on Local Moran's $R$ to identify significant interaction spots, where the variance for local $R_i$ is derived as:

$$Var(R_i) = 2\frac{(n-1)^2}{n^2}\sigma_1^2\sigma_2^2 \sum_{j=1}^{n} w_{ij}^2 + 2\frac{(n-1)^2}{n^2}\sigma_1^2\sigma_2^2 w_{ii}^2 \qquad (7)$$

where $\sigma_1$ and $\sigma_2$ are the standard deviations for ligand and receptor, respectively (see more details in Supp Note 1).

To avoid picking interacting spots with low sender signals and low receiver signals in the neighbourhood, which would result in a high positive Local Moran's $R$, we adapted to the quadrant method of Moran's $I$ and refined the significant spots to be those with higher-than-average level for either sender signals or receiver signals, i.e. Local $p_i = 1$ when $x_i - \bar{x} \leq 0$ and $y_i - \bar{y} \leq 0$.

## Simulation

The simulation approach was adapted from SVCA[20] and was based on Thrane's melanoma dataset with 293 spots[29]. In SVCA, the variance of each gene was decomposed using a multivariate normal model into the intrinsic factor which can be inferred from expression patterns of all other genes, the environmental factor which can be imputed from spatial adjacency, the noise factor, and most importantly, the interaction factor which is a linear combination of neighbour cell expression profiles. After fitting the model to real spatial data, SVCA rescales the interaction factor to simulate different degrees of interaction. Here, with the hypothesis that genes correlate more with binding partners instead of all other genes, we adapted SVCA by replacing the intrinsic factor modelled from all genes with corresponding receptor subunits for each ligand gene. Please refer to SVCA for detailed protocols[20]. SVCA settings were kept except the term X which was the expression profile across all spots of all genes except the molecule of interest (dimensions = the number of molecules −1), and adapted as the expression profile across all spots only on the corresponding receptor genes (dimensions = the number of receptor subunits). Briefly, the adapted SVCA model was fitted for each ligand gene in the

ligand–receptor database using maximum likelihood. The cell–cell interaction covariance was then rescaled to simulate circumstances of no interaction (0%), 25% interaction, 50% interaction, 75% interaction, and 99% interaction. For negatively correlated pairs observed from all scenarios except 0% interaction, we reversed the signs for each simulated ligand expression value.

## Comparison with other models

Given limited methods serving the exact same functions to identify ligand-receptor interactions directly from spatial omics, 4 methods with limited degrees of overlap were included in the comparison despite unfavourable simulation settings. We applied SpatialDM (both approaches, non-single cell resolution, $l = 1.2$, cut-off=0.2), CellChat (v.1.1.3; default trimean setting and truncatedMean with trim = 0), Giotto (v.1.0.4; default setting), SpaTalk (v.1.0; loss option changed to mse), and SpatialCorr (v.1.1.0; default setting) to the positive-interaction simulations to compare the true positive rate (TPR), and to the no-interaction simulation to compare the false positive rate (FPR). As CellChat and Giotto results were presented on a cluster level, we kept the lowest $p$-value for each ligand-receptor pair across all cluster-cluster results. Receiver operating characteristic (ROC) was plotted for each method, and Area Under ROC (AUROC) were compared under each interaction scenario. We also compared the computation time for 1000 LR pairs of the aforementioned methods. Given the high computation efficiency of SpatialDM, 1 core was applied for the run time. We run all other methods using 50 cores except SpaTalk and SpatialCorr. The number of spots was varied from 1000 to 10,000 (1 million for the $z$-score approach of SpatialDM).

We also applied different models in the melanoma dataset with the aforementioned settings. In addition, we shuffled the curated ligand-receptor to generate a 663-pair negative control list. In theory, interactions between these LR pairs were not documented before. We applied SpatialDM and the aforementioned methods with the same settings on the negative control list for FPR comparison.

## Experimental datasets and processing settings

Three datasets of different sizes and from different sequencing platforms were used to showcase the framework, including (1) Thrane's melanoma dataset (sample 1 rep 2, 293 spots, ST[29]), (2) All intestine samples probed by Visium from Corbett et al., containing 8 slices from 3 time points and 4 donors, respectively[7] and (3) a SVZ sample (FOV of 5) slice from Eng's[48]. We mainly showcased the permutation approach in the melanoma and SVZ datasets (Global: FDR < 0.1, Local: $p$-value < 0.1), and the $z$-score approach in the intestine datasets (Global: FDR < 0.1, Local: $p$-value < 0.1).

**Cell type annotation.** For the melanoma dataset, scRNA-seq and marker gene lists of each of the 7 cell types were publicly available[33]. Cell type composition in each spatial transcriptome spots were computed using RCTD v.2.0.1[30] based on the spot mRNA expression and the marker gene list. For the intestine and SVZ datasets, we directly used cell type annotations from the original study[7].

**Verification in scRNA-seq.** Dimension reduction was performed in scRNA-seq using tSNE. Cell-type annotations were performed by Tirosh, et al.[33]. FCER2 and CR2 expressions were visualised in tSNE and violin plots.

## Additional utility analyses in SpatialDM

**Histology clustering of significant pair using SpatialDE.** SpatialDE which was originally invented to distinguish and classify genes with spatial patterns of expression variation with its automatic expression histology module (SpatialDE.aeh), enabling expression-based tissue histology. We simply re-implemented `SpatialDE.aeh.spatial_patterns` function by feeding the local Moran statistics to

cluster all selected interactions into 3 (in melanoma) or 4 (in intestine) patterns. The input here is the binary matrix of either local permutation or z-score selected spots (0 for non-significant spots, 1 for selected spots). Alternatively, other local statistics like local $R_i$ can serve as SpatialDE input to explore interaction-level histology.

**Pathway enrichment.** For selected pairs, we counted the number of pairs belonging to each pathway as documented in CellChatDB v.1.1.3 which was visualised in the dot plot x-axis. We also computed the percentage of the pairs in relation to all pairs belonging to the respective pathway in the dataset. Notably, Fisher's exact test calculates the probability that the association between the queried interactions and the interactions belonging to a given pathway occurs purely by chance, which is indicated by the dot size.

**Chord diagram.** Chord diagram has been implemented in many ligand-receptor interaction packages. On the one hand, SpatialDM allows the identification of interactions for each spot without spatial or biological clustering. On the other hand, it is useful to integrate cell type information when interpreting the results. We include the utility based on HoloView[49] to visualise the interacting cell types, based on each spot's Moran statistics and cell type decomposition value. By running `pl.chord_celltype` for a selected pair, the relative edge number for a cell type pair

$$n_{AB} = \sum_{i,j} w_{ij} R_{i,sender} A_i R_{j,receiver} B_j, \qquad (8)$$

where A, B denote 2 independent cell types from the annotation, respectively. We also provide the function `pl.chord_celltype_allpairs` to aggregate $n_{AB}$ along all cell type combinations. In addition, given a cell type combination AB, the selected interactions can be visualised using `pl.chord_LR` in a similar fashion where the relative edge number for an interaction

$$n_k = \sum_{i,j} w_{ij} R_{k,sender} A_i R_{k,receiver} B_j, \qquad (9)$$

**Differential analyses.** Colon samples (A1, A2 for adult, A3, A4, A8 and A9 for foetus) and their Global Moran z-scores (1,486 pairs) were extracted for differential analyses. If either ligand or receptor was not detected in a sample, the z-score was forced to 0. For each pair, linear regression models were fitted to the 6 z-scores twice, with (full model) or without (reduced model) condition information. A likelihood ratio test was performed to calculate the p value for differential communication. Specifically, the difference between log-likelihood from the full vs. reduced models was then subjected to Chi-Squared test for the differential p-values[50].

**Correlation between local Moran statistics and cell type weights**
We fitted the linear model on the local Moran p-values computed by SpatialDM ($N \times k$) to predict cell-type results ($N \times m$, N: number of spots, k: number of selected interactions, m: number of cell types, decomposition results were performed using RCTD in the melanoma data or by the authors in the intestine data). All data were used to train the linear model and for testing. Pearson's R was then computed by comparing the predicted decomposition results with the real ones (both of $N \times m$ shapes).

**Fine tune with auto-correlation weights**
With a hypothesis that spatially significant pairs will have a certain degree of auto-correlation for the ligand or receptor, we integrated ligand/receptor Moran's R in simulated data.

Auto-correlation Moran's $I_l$ (ligand) and $I_r$ (receptor) are defined as:

$$\begin{aligned} I_l &= \frac{\sum_i \sum_j w_{ij}(x_i - \bar{x})(x_j - \bar{x})}{\sum_i (x_i - \bar{x})^2} \\ I_r &= \frac{\sum_i \sum_j w_{ij}(y_i - \bar{y})(y_j - \bar{y})}{\sum_i (y_i - \bar{y})^2}, \end{aligned} \qquad (10)$$

where x denotes ligand expression, y denotes receptor expression. The fine-tuned $R = w_l * I_l + w_r * I_r + R_{lr}$. We used $w_l = 0.17$ and $w_r = 0$ in the simulation data (learned from a logistic regression on a separate dataset) and used $w_l = 0$, $w_r = 0$ for all experimental datasets.

## Reporting summary
Further information on research design is available in the Nature Portfolio Reporting Summary linked to this article.

## Data availability
All datasets used here are previously published and publicly available (Raw mRNA counts, log-transformed mRNA counts, and spatial coordinates of the melanoma data were obtained from https://github.com/msto/spatial-datasets, ; Raw mRNA counts and spatial coordinates of the intestine data were obtained from https://simmonslab.shinyapps.io/FetalAtlasDataPortal/, GEO: GSE158328 [https://www.ncbi.nlm.nih.gov/geo/query/acc.cgi?acc=GSE158328], Raw mRNA counts and spatial coordinates of the SVZ data were obtained from https://github.com/CaiGroup/seqFISH-PLUS). Source data are provided with this paper. For easier reuse, we also included them in the SpatialDM Python package and figshare tabs as follows, the melanoma data:*spatialdm.datasets.melanoma*(), figshare tab:*mel_adata*the intestine data (e.g. A1):*spatialdm.datasets.A*1, figshare tab:*A*1, same for*A*2,*A*3,*A*4,*A*6,*A*7,*A*8, the SVZ data:*spatialdm.datasets.SVZ*(), figshare tab:*SVZ*. The ligand-receptor databases are available from CellChat repository (https://github.com/sqjin/CellChat/tree/master/data). Source data are provided with this paper.

## Code availability
SpatialDM is an open-source Python package freely available at https://github.com/StatBiomed/SpatialDMand https://doi.org/10.5281/zenodo.7920811[51]. We make it convenient by directly integrating with Scanpy or Anndata objects. Detailed documentation and the analysis notebooks to reproduce results in this paper are also included in this repository (https://spatialdm.readthedocs.io/). All data analysed in the paper are available through the figshare linkhttps://doi.org/10.6084/m9.figshare.22960949.

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

## Acknowledgements

We thank Rio Sugimura, Martin Cheung and Langqi Gong for biological insights on discussing melanoma analyses and Chen Qiao for technical discussion on ST data modelling. Corbett et al. kindly provided the full list of their identified LRIs for us as a reference. We also thank Shoufa

Chen and Mingze Gao for helping with the Python implementation and the package-releasing process. This project is supported by Innovation Technology Commission Funding (Health@InnoHK), GRF (17126421), MOST Key Project (2022YFA1105400), NSFC/RGC (CRS_HKU703) (P.L.), and the University of Hong Kong through a startup fund and a seed fund (Y.H.). Z.L. is supported by Presidential Scholarship of the University of Hong Kong.

## Author contributions

Y.H. and P.L. conceived and supervised the study. Z.L. and Y.H. designed the project. Z.L. implemented the SpatialDM package and performed all data analysis, with support from T.W. and Y.H.; T.W. derived the analytical null distributions. Z.L. and Y.H. wrote the manuscript with inputs from all authors.

## Competing interests

The authors declare no competing interests.
