## [Peer Review File · Nature Communications]

SpatialDM for rapid identification of spatially co-expressed ligand-receptor and revealing cell-cell communication patternsReviewer #1 (Remarks to the Author):

In this manuscript, Li et al. described SpatialDM, which is a statistical method utilizing bi-variant Moran's I statistic to detect spatially interacting receptor-ligand pairs and locations of significant interactions. SpatialDM is presented as a "toolbox" to perform additional downstream analyses like using SpatialDE (Nature Methods, 2018) to cluster spots that have significant local interactions and obtain interaction "patterns".

The manuscript is generally well written and reports biologically plausible results from real datasets. Our major concerns, as discussed below, are about insufficient benchmarking and a lack of clarity about the local statistic.

1. Insufficient benchmarking - it is surprising that the authors did not include scHOT (Nature Methods, 2020) in their benchmarking or comparative discussions on why SpatialDM's formulation should be regarded as more appropriate, although scHOT accomplishes nearly identical goals as SpatialDM.

2. The local statistic: Eqn 3 in methods is questionable. It seems to be an adoption of Luc Anselin's original formulation for local autocorrelation based on Moran's I, but it is unclear why: (a) there isn't any normalization term and (b) if simply adding these two terms can yield high sensitivity and specificity. Again, this needs a separate simulation study.

3. Dealing with multimeric complexes - it is questionable to take the average values of the expression levels of genes that encode proteins for multimeric complexes. I think CellPhoneDB might have taken a similar approach but given that unnormalized gene expression levels should not be compared, especially to summarize their co-expression, the authors actually need a separate simulation study for this.

Reviewer #2 (Remarks to the Author):

SpatialDM provides a scalable and efficient solution for the analysis of ligand-receptor co-expression patterns in space. The tool is well-made and the proposed applications are representative of why it provides a needed solution to the analysis of Spatial Transcriptomics data.

However, the text of the manuscript and the quality of the figures need more work. In particular, the text is often insufficient to highlight the reasons why certain steps are done, and it would be unclear to a reader without extensive background in CCC. There is also a need for the analyses downstream of the global/local Moran's I calculations to be polished a bit further. Specifically:

Major Comments:

- In their modest benchmark, the authors say that their method is "substantially outperforming" which is just unnecessary in such a limited benchmark setting, we suggest that the authors rephrase this. Moreover, in the same setting CellChat's two means vary substantially in performance, yet while Trimean is expected to be more conservative, such a difference between two fairly similar functions is unexpected. Have the authors checked if they have enough observations for CellChat's Trimean, or if an unexpected artefact is skewing the results?

- Also, while the benchmark is a good addition, "As CellChat and Giotto results were presented on a cluster level, we kept the lowest p-value for each ligand-receptor pair across all cluster-cluster results" simply suggests that these methods are not directly comparable to SpatialDM. Thus, we recommend that the authors deemphasize the comparison with these methods, or explicitly highlight that the proposed setting is likely biased for their method.

• We compliment the authors on providing a manuscript with a concise and clear message. However, at the same time, the text is often insufficient to explain why/how certain steps for the analyses are done. For example, the authors use SpatialDE as a way to obtain spatially-variable LR patterns. Yet, besides mentioning that local I is treated as 0 or 1 for the analysis, a proper

explanation for the use of binarized statistics, and not continuous, the assumptions of the analysis, and SpatialDE itself are lacking. We thus ask the authors to extend their methods section with some of the missing detail.

- “Indeed, we found that the local interaction scores are good predictors of the cell types (Pearson’s $R=0.928$; linear regression; Extended Data Fig. 2B)”. This result seems intriguing, though it is unclear how the linear model is fit? Could the authors explain and extend this analysis?
- On the same note, can the authors also provide a functionality to obtain a local I summary per LR/pattern across niche/cell type? The pie charts go in that direction, but are rather crude. We refer the authors to Niches (<https://www.biorxiv.org/content/10.1101/2022.01.23.477401v2.full>) and scriabin (<https://www.biorxiv.org/content/10.1101/2022.02.04.479209v3.full>) as source for potential ideas.
- An application to FISH-based transcriptomics dataset would show the broadly applicability of SpatialDM - Visium is not the only technology available, and a good spatial method should be applicable to multiple technologies, which have in this case quite distinct features.
- Counting the number of interactions per pathway is perhaps too simplistic. A simple alternative could be a one-sided Fisher’s Exact test?

Minor Comments:

- It would be of interest to get a better understanding of what is the minimum number of spots in which we can use the analytical solution and obtain a decent correlation with permutations (e.g. an arbitrary 0.90 Spearman Coeff)
- We compliment the authors for the effort put in the implementation, documentation and tutorials for SpatialDM. We also appreciate computational scalability as a major advantage of SpatialDM, thus a critique of the current implementation would be that a pandas dataframe is used to store the typically sparse matrices, which is bad practice in terms of scaling for RAM. Would it be possible to change SpatialDM to work directly on an adata object, or alternatively with a sparse matrix, and avoid the conversion to a dataframe?
- The figures are generally of low quality, with prime example being Supp Figure 2E which is difficult to interpret. Also, many alternative ways to perform and visualise an enrichment test exist, so please consider improving the quality of this figure and the quality of figures in the manuscript all together.
- It would be informative to see how results change if different sources of ligand-receptor knowledge were used
- An extended explanation how z-scores are calculated across contexts would be appreciated.
- The algebraic mean is used to account for the expression of complexes, but how do the authors deal with 0s? If one subunit is not expressed, then the complex cannot be active.
- A general discussion of the limitations of using a bivariate Moran’s I is missing. For example, is it capable of dealing with negative abundances? Does it consider pleiotropy between ligands? Also, in general does the spatial co-expression of ligand-receptors necessarily translate into CCC events?

Reviewer #1:

1. Insufficient benchmarking - it is surprising that the authors did not include scHOT (Nature Methods, 2020) in their benchmarking or comparative discussions on why SpatialDM's formulation should be regarded as more appropriate, although scHOT accomplishes nearly identical goals as SpatialDM.

Response: We appreciate the reviewer for suggesting scHOT. We integrated scHOT in our introduction (page 2, highlighted), as scHOT method is indeed relevant and robust in identifying coordinated changes in spatial context or along a development trajectory. But scHOT substantially differs from SpatialDM in the method and objectives. scHOT aims to detect gene pairs that have significantly differential correlations (in local regions), compared to a general overall correlation. Specifically, it calculates the local correlation via a standard weighted Spearman correlation for each spot and by default it tests if the observed standard deviation of the local correlation vectors is extreme compared to a null distribution by permuting cell locations.

However, SpatialDM aims to detect gene pairs (ligand-receptor pair) that have spatial co-expression (i.e., spatial correlation), namely if ligand and receptor are co-expressed within a reasonable geographical distance. Specifically, the Moran's R statistic in SpatialDM uses the ligand in neighbor spots for the receptor in a certain spot (and vice versa), while weighted correlation methods like scHOT always use the gene pair values in the same spots.

Therefore, scHOT is for detecting spots with differential co-expression (correlation) across space but may not be able to detect features with spatial co-expression.

Nonetheless, we agreed that scHOT could be a good counterpart to compare, so we have performed scHOT on our simulated data. Surprisingly we found its differential correlation statistic (or p-value) is a poor indicator of spatial co-expression despite controlling a low false positive rate (see Fig. 1R below). As clarified above, this may be caused by the different purpose of the testing methods.

Figure R1. Simulation results with scHOT. The left panel is in the same format as main Figure 1f, and the right panels are ROC curves for the same simulation data in main Figure 1g and 1h.

To further confirm that the variability of locally weighted correlation is not a good indicator for detecting (global) spatial correlation, we also include the comparison with SpatialCorr (Bernstein et al, Cell Reports Methods, 2022), which replaced the weighted Spearman correlation with weighted Pearson's correlation for more efficient computing with WR-test.

Interestingly, we found the SpatialCorr returns more positive results than scHOT, while it is still not powerful enough to detect the spatial co-expression compared to SpatialDM (main Figure 1f-h).

2. The local statistic: Eqn 3 in methods is questionable. It seems to be an adoption of Luc Anselin's original formulation for local autocorrelation based on Moran's I, but it is unclear why: (a) there isn't any normalization term and (b) if simply adding these two terms can yield high sensitivity and specificity. Again, this needs a separate simulation study.

Response: Thank the reviewer for the useful feedback; we assume it is the Eqn 6 that the reviewer mentioned.

Here, we have updated the gene-wise normalization in the package and have performed additional simulations to illustrate the specificity.

Initially, we didn't do spot-wise normalization for each gene, given that the gene pair of interest is fixed in local Moran statistics. Now we realized even though the normalization does not affect the per-gene analysis, it can be helpful for gene pairs between the ligand and the receptor and among subunits, thanks to the helpful feedback. In the updated local statistics, we did a gene-wise normalization by dividing the gene maximum, (optionally) followed by scaling the same way as in Scanpy (highlighted under Method -- Significant interaction spots).

Also, we agreed that a separate simulation to demonstrate the effectiveness of the local co-expression can be valuable. However, it is generally challenging to generate a full dataset with preserving the local patterns. Therefore, we only added simulations to assess the false-positive control but not the power analysis. Nonetheless, we found with the permuted data, the detected co-expressed spots are significantly lower than the observed results in the real data. In melanoma, a median of 11 spots (permuted) vs 22 spots (real) were selected out of 293 with permutation approach ($p < 0.1$, no. pairs=103); in intestine A2, a median of 6 spots (permuted) vs 37 (real) spots selected with the analytical approach ($p < 0.1$; no. pairs=442), which suggests the potential effectiveness of our method in detecting the locally co-expressed spots. Considering that the primary goal of our manuscript is to detect spatially co-expressed ligand-receptor at a global level (i.e., feature selection), we hope it is acceptable with this imperfect simulation in demonstrating the local spot detection. We have discussed this limitation in the discussion section (page 14) and anticipate future simulation works will better facilitate this demand.

Figure R2. The distribution of the number of detected local spots with $p < 0.1$ in both real observations and after the spots are permuted (Left: melanoma, right: intestine A2).

3. Dealing with multimeric complexes - it is questionable to take the average values of the expression levels of genes that encode proteins for multimeric complexes. I think CellPhoneDB might have taken a similar approach but given that unnormalized gene

expression levels should not be compared, especially to summarize their co-expression, the authors actually need a separate simulation study for this.

Response: We have addressed the normalization issue under the previous comment. We have previously relied on the arithmetic mean when dealing with multimeric complexes, as current spatial transcriptome data still face low sequencing depth, meaning that value zero seen in one subunit may be caused by technical drop-out especially when the other subunit(s) is non-zero. However, we reckon some analysts may prefer a more conservative approach to averaging (e.g., to take geometric means). We have included geometric means as an alternative when researchers wish to take zero expressions of any subunits as zero expressions of the whole protein complex.

The geometric mean does generate more stringent results than the arithmetic mean. Take the seqFISH+ dataset as an example, the former only picked out 2 pairs, whereas the latter selected 23 pairs (FDR < 0.1, permutation). Pairs expressing multi-subunits cannot be detected by geometric mean. See the examples below.

Figure R3. Two exemplar pairs selected only by arithmetic mean. Both compose multimeric receptors.

Reviewer #2:

Major Comments:

1. In their modest benchmark, the authors say that their method is “substantially outperforming” which is just unnecessary in such a limited benchmark setting, we suggest that the authors rephrase this. Moreover, in the same setting CellChat’s two means vary substantially in performance, yet while Trimean is expected to be more conservative, such a difference between two fairly similar functions is unexpected. Have the authors checked if they have enough observations for CellChat’s Trimean, or if an unexpected artefact is skewing the results?

Response: Thank the reviewer for the constructive suggestions. We have rephrased the comparisons and pointed out the limitations of our simulations wherever appropriate in the manuscript (e.g. page 6, 17, highlighted) and clarified CellChat’s discrepancy as below.

As mentioned in CellChat GitHub tutorials (link below), Trimean approximates 25% truncated mean. It implies that a gene is regarded as not expressed when it is expressed in fewer than 25% of spots in a group. Given the sparsity of spatial transcriptome and specificity of some ligand/receptor expression, gene expression in fewer than 25% of spots is ubiquitous. Therefore, Trimean results have limited power in any simulations. In their recent tutorial, the authors also recommended TruncatedMean over Trimean for spatial transcriptomic data.

Therefore, we have excluded Trimean in main analysis of the manuscript (highlighted, page 6, Fig. 1H).

Ref: Jin et al. (2020). CellChat. Retrieved from <https://htmlpreview.github.io/?https://github.com/sqjin/CellChat/blob/master/tutorial/>

2. Also, while the benchmark is a good addition, “As CellChat and Giotto results were presented on a cluster level, we kept the lowest p-value for each ligand-receptor pair across all cluster-cluster results” simply suggests that these methods are not directly comparable to SpatialDM. Thus, we recommend that the authors **deemphasize the comparison with these methods**, or explicitly highlight that the proposed setting is likely biased for their method.

Response: We totally agree with the comment. On page 6, we pointed out that all methods in comparison may not be favoured by the simulation setup with the objective to capture spatially co-varying ligand-receptor interactions (highlighted), especially for those aiming for detecting communicating cells instead of features.

3. We compliment the authors on providing a manuscript with a concise and clear message. However, at the same time, the text is often insufficient to explain why/how certain steps for the analyses are done. For example, the authors use SpatialDE as a way to obtain spatially-variable LR patterns. Yet, besides mentioning that local I is treated as 0 or 1 for the analysis, **a proper explanation for the use of binarized statistics**, and not continuous, the assumptions of the analysis, and **SpatialDE itself** are lacking. We thus ask the authors to extend their methods section with some of the missing detail.

Response: Thank you for the great suggestion. Indeed, some details are lacking in the original manuscript. Now, we have substantially extended the methods sections (p.18). SpatialDE, originally invented to distinguish and classify genes with spatial patterns of expression variation with its automatic expression histology module (SpatialDE.aeh), enables expression-based tissue histology. To cluster all selected interactions, we simply re-implemented SpatialDE.aeh.spatial patterns function by feeding the binary local significance (0 for non-significant spots, 1 for selected spots). Alternatively, we can feed local Moran's R itself as normalization has been performed in the updated version, which achieved similar effects, especially on the coarse patterns.

Figure R4. SpatialDE clustering of different local Moran statistics. Left: Moran's R as input; right: binary significance input.

4. “Indeed, we found that the local interaction scores are good predictors of the cell types (Pearson’s $R=0.928$; linear regression; Extended Data Fig. 2B)”. This result seems intriguing, though it is unclear how the linear model is fit? Could the authors explain and extend this analysis?

Response: We apologize for missing the relevant Methods and have added more details on page 19. Briefly, we fitted the linear model on the local Moran p -values computed by SpatialDM ($N \times k$) to predict cell-type results ($N \times m$, N : number of spots, k : number of selected interactions, m : number of cell types, decomposition results were performed using RCTD in the melanoma data or by the authors in the intestine data). All data were used to train the linear model and for testing. Pearson’s R was then computed by comparing the predicted decomposition results with the real ones. The prediction could be influenced by the decomposition results and training/testing size.

We also extended such predictions to Random Forest model, where the cross-validation ($cv=5$) generated a Pearson correlation of 0.866.

5. On the same note, can the authors also provide a functionality to obtain a local I summary per LR/pattern across niche/cell type? The pie charts go in that direction, but are rather crude. We refer the authors to Niches (<https://www.biorxiv.org/content/10.1101/2022.01.23.477401v2.full>) and scriabin (<https://www.biorxiv.org/content/10.1101/2022.02.04.479209v3.full>) as source for potential ideas.

Response: Thanks to the recommendations by the reviewer especially for the figures in Scriabin, we have replaced the pie charts with more illustrative chord diagrams (functions: `pl.chord_celltype`, `pl.chord_LR`, and `pl.chord_celltype_all`), in which the nodes are the signalling cell types and the edge colours indicate the direction of signalings (same colour with source cell types). We use the chord diagram to replace the pie charts (e.g., Fig. 2D, Fig. 3D), and included the Method details on page 19.

6. An application to FISH-based transcriptomics dataset would show the broadly applicability of SpatialDM -

Visium is not the only technology available, and a good spatial method should be applicable to multiple technologies, which have in this case quite distinct features.

Response: We appreciate the reviewer’s suggestion. Given the low throughput of most FISH data, the starting LR pairs are generally low (see Table below), hence making it less appropriate for the ligand-receptor analysis, except reliable imputation which is generally challenging (see Chen et al Bioinfo 2021; PMID: 34252941). On the other hand, the seqFISH+ dataset has a high number of starting interactions, so we have performed in-depth

analyses of the dataset (Extended Data Fig. 1), where we detected 23 significant interaction pairs.

Additionally, we also applied SpatialDM to a recent technology Stereo-seq and obtained reasonable number of significant ligand-receptor pairs. Taken together, these results suggest the potential of wide applicability of our method.

Dataset	no_genes	no_interactions (in CellChatDB)	No_significant_pairs (z-score, FDR<0.1)	reference
osmFISH	33	0	0	[1]
merfish	155	16	6	[2]
starmap	1020	28	9	[3]
seqFISH+	10000	1157	23	[4]
Stereo-seq	28579	1884	1236	[5]

[1] Codeluppi S., Borm L.E., Zeisel A., La Manno G., van Lunteren J.A., Svensson C.I., Linnarsson S. Spatial organization of the somatosensory cortex revealed by osmFISH. *Nat. Methods*. 2018; 15:932–935.

[2] Moffitt J.R., Bambah-Mukku D., Eichhorn S.W., Vaughn E., Shekhar K., Perez J.D., Rubinstein N.D., Hao J., Regev A., Dulac C. et al. . Molecular, spatial, and functional single-cell profiling of the hypothalamic preoptic region. *Science*. 2018; 362:eaau5324

[3] Wang X., Allen W.E., Wright M.A., Sylwestrak E.L., Samusik N., Vesuna S., Evans K., Liu C., Ramakrishnan C., Liu J. et al. . Three-dimensional intact-tissue sequencing of single-cell transcriptional states. *Science*. 2018; 361:eaat5691.

[4] Eng C.H.L., Lawson M., Zhu Q., Dries R., Koulouina N., Takei Y., Yun J., Cronin C., Karp C., Yuan G.C. et al. . Transcriptome-scale super-resolved imaging in tissues by RNA seqFISH+. *Nature*. 2019; 568:235–239.

[5] Chen, Ao, et al. "Spatiotemporal transcriptomic atlas of mouse organogenesis using DNA nanoball-patterned arrays." *Cell* 185.10 (2022): 1777-1792.

7. Counting the number of interactions per pathway is perhaps too simplistic. A simple alternative could be a one-sided Fisher's Exact test?

Response: Thanks to the reviewer's recommendation, we adapted one-sided Fisher's Exact test for pathway enrichment analyses. In the updated dot plot, dot size now denotes the $-\log(p)$ from Fisher's Exact test and the dot color is still related to the percentage of significant interactions over all interactions in a certain pathway. Examples are Fig. 2C, Fig. 3E and Fig. 4C-D.

Minor Comments:

- It would be of interest to get a better understanding of what is the minimum number of spots in which we can use the analytical solution and obtain a decent correlation with permutations (e.g. an arbitrary 0.90 Spearman Coeff)

Response: I appreciate this very good comment. We performed a down-sampling test from the mouse organogenesis dataset generated with Stereo-seq from 10k spots down to 100 spots (Chen et al, *Cell* 2022. PMID: 35512705). We observed that SpatialDM is generally robust to reduced number of spots, returning high Spearman Coefficients across most scenarios (e.g., $R=0.902$ with 2000 spots; See figure below).

Figure R6. Spearman correlation coefficient bar plot for downsampled dataset.

- We compliment the authors for the effort put in the implementation, documentation and tutorials for SpatialDM. We also appreciate computational scalability as a major advantage of SpatialDM, thus a critique of the current implementation would be that a pandas dataframe is used to store the typically sparse matrices, which is bad practice in terms of scaling for RAM. Would it be possible to change SpatialDM to work directly on an adata object, or alternatively with a sparse matrix, and avoid the conversion to a dataframe?

Response: Thank you for the valuable feedback. We have now implemented it fully based on sparse matrix and supporting adata object as the default input.

- The figures are generally of low quality, with prime example being Supp Figure 2E which is difficult to interpret. Also, many alternative ways to perform and visualise an enrichment test exist, so please consider improving the quality of this figure and the quality of figures in the manuscript all together.

Response: Thank the reviewer for pointing out the figure quality issue. For Supp Figure 2E, we have replotted it with goatools. The takeaway from the GO results remains the same— CD23 selected B cells are more likely to play an anti-tumour role in the melanoma, as positive regulation of T cells and macrophages are enriched among all biological functions. (Figure Supp Fig. 3E; also below). For other plotting, we have also improved its presentation thanks to the reviewers' suggestions, including replacing the pie charts with chord diagram, and upgrading the pathway dot plot with Fisher's test results.

Figure R7. GO (biological function) result for genes enriched in CD23 selected spots.

- It would be informative to see how results change if different sources of ligand-receptor knowledge were used

Response: Thank you for the constructive suggestions. We are not entirely sure if the different sources mean different input ligand-receptor databases or different interaction source patterns among ligand-receptor pairs. Nonetheless, both are very interesting aspects to explore, so we performed analyses for both.

Firstly, we have compared CellChatDB (1940 pairs) with iCellnetDB (1034 pairs) in all 8 intestine samples. Although the overlaps (3–400 pairs) show generally aligned results, many more pairs were selected when using CellChatDB (See Table). Given SpatialDM's good control over false positives, we recommend using an input database as complete as possible.

	A1	A2	A3	A4	A6	A7	A8	A9
CC_all	987	1050	1083	1147	1095	1056	1095	1090
CC_selected	383	447	442	613	532	472	311	335
icellnet_all	567	595	620	659	629	583	618	600
icellnet_selected	234	264	249	378	338	295	173	192

Secondly, we separated all ligand-receptor pairs into long-distance (secreted signalling) and short-distance (cell-cell contact and ECM). Calculations remained similar for the former, while the signalling distance was restricted to max 1 spot radial distance for the latter. We have implemented and set it as default. See the revised Methods (page 15) and updated results (minor alterations highlighted in green throughout the results).

- An extended explanation how z-scores are calculated across contexts would be appreciated.

Response: For both global and local R, z-scores are defined as (global or local) Moran's R divided by the standard deviation (variance squared root), the equation for each contained in Methods. For global, the calculation of Moran's R is specified in Eq (1), and the variance is written in Eq (5). Local Moran's R is specified in Eq (6), and local variance is specified in Eq (7). More details are included in the Supplementary Notes. When the context (dataset) changes, users only need to customize parameters in the weight matrix calculation in Eq (2), as we have demonstrated the broad applicability of z-scores in large and small dataset under the first minor comment.

- The algebraic mean is used to account for the expression of complexes, but how do the authors deal with 0s? If one subunit is not expressed, then the complex cannot be active.

Response: Thanks for raising this issue too. Please see our response to Reviewer 1, point 3.

- A general discussion of the limitations of using a bivariate Moran's I is missing. For example, is it capable of dealing with negative abundances? Does it consider pleiotropy between ligands? Also, in general does the spatial co-expression of ligand-receptors necessarily translate into CCC events?

Response:

Thank the reviewer for this insightful comment. We agreed the interpretation from bivariate Moran's I was limited in our original version, but the comments enlightened us to expand the Discussion. In paragraph 3 of Discussion, we discussed potential improvement of the bivariate Moran's I like using spatial proteomic or with greater resolution. Application of SpatialDM on such technically-advanced data will translate to CCC insights more directly. As for negative abundances, SpatialDM can deal with them we take demean of the expression. One example is our simulated data.

The pleiotropy between ligands is often caused by different ligands binding to the same receptors. If multiple ligands compete for a common receptor in most cases, many of the conclusions from SpatialDM will be less interpretable. However, we learned from the 2 findings below that common receptors can accurately sense different ligands of different concentrations simultaneously. Therefore, pleiotropy may not have strong impact to our current pipeline and results. Nonetheless, we agreed that this could be a potential issue and we mentioned in the discussion (p.14).

Ref: Kirby D, Rothschild J, Smart M, Zilman A. Pleiotropy enables specific and accurate signaling in the presence of ligand cross talk. *Phys Rev E*. 2021 Apr;103(4-1):042401. doi: 10.1103/PhysRevE.103.042401. PMID: 34005921.

Singh V, Nemenman I (2017) Simple biochemical networks allow accurate sensing of multiple ligands with a single receptor. *PLOS Computational Biology* 13(4): e1005490. <https://doi.org/10.1371/journal.pcbi.1005490>

Reviewer #1 (Remarks to the Author):

The authors have incorporated our comments. There are a few remaining concerns, which we believe are easy to address. Addressing point (a) below will add more to the rigor of their claims.

(a) Fig 3a: the high Pearson values are likely driven by the few outliers in each case. Looking at the general cloud of points, it doesn't seem that the Pearson values would be as high as the reported ones. The authors need to comment on this. Are receptor/ligand expressions spatially congruent across samples?

(b) The text needs a thorough proofreading (e.g., after Eq 6, the word "or" appears twice)

(c) The authors need to improve the visualizations (e.g., the spots in Fig 2d and 2f should be as easy to discern as the spots in Fig 2c. In Fig 2b, why are there some white marks in the RCTD results?)

Reviewer #2 (Remarks to the Author):

We appreciate the effort that the authors put into revising the manuscript and adapting the tool to best coding practices. However, we still have a few remaining comments.

Major comments:

- We noticed that while the global scores are bound between -1 to +1, the local spatialDM scores range between $-\infty$ to $+\infty$, is this something that is intended and anticipated? We also point the authors to the point of the other reviewer regarding the issue with the normalization term from the local univariate equation. This should not be confused with a priori normalization of the expression.

- Given the large-scale and still ongoing refactoring of the code, have the authors made sure that their initial results are reproducible and the code works? A github link to the analyses presented with versioning is necessary to ensure best reproducibility practices. Importantly, currently SpatialDM and its tutorial do not work.

Minor comments:

- In their last response the authors mention that "spatial proteomic or with greater resolution" would improve SpatialDM. We are not sure how proteomics is related to the inherent limitations to SpatialDM specifically, rather the authors should address that the normalization of the local variables is based solely according to the mean of each variable, thus it does not e.g. necessarily encode the variation within each variable, etc. In other words, authors should discuss limitations specific to their method, not the field.

Reviewer #1 (Remarks to the Author):

The authors have incorporated our comments. There are a few remaining concerns, which we believe are easy to address. Addressing point (a) below will add more to the rigor of their claims.

(a) Fig 3a: the high Pearson values are likely driven by the few outliers in each case. Looking at the general cloud of points, it doesn't seem that the Pearson values would be as high as the reported ones. The authors need to comment on this. Are receptor/ligand expressions spatially congruent across samples?

Response:

(1) Thanks for raising the point. Pearson's correlation coefficient may indeed be affected by outliers. Here, we further calculated Spearman's correlation to mitigate the impact of outliers (Fig R1 & Updated Fig. 3a). As expected, the correlation coefficient is indeed lower than Pearson's (from 0.936 to 0.712 for A1 sample). On the other hand, we argue that those outliers (with high Moran's R values) are the communicating ligand-receptor pairs, hence at a certain perspective it is beneficial that the metric has more emphasis on them. Nonetheless, we keep both types of correlations in this revision to avoid potential confusion (Updated Fig. 3a). Furthermore, the comparison results are consistent between Pearson's and Spearman's, namely high correlations between slices from the same sample versus low correlations among slices from different samples (p10, highlighted).

Fig R1. Similar to Figure 3a, except the correlation coefficients labelled on the top left are Spearman's correlation coefficient.

(2) From our understanding of the second question on whether “receptor/ligand expressions spatially congruent across samples”, the short answer is yes. Within one sample, under the null we assume it is spatially i.i.d., and follows a normal distribution. Across samples, we make them comparable by normalizing them to z scores.

(b) The text needs a thorough proofreading (e.g., after Eq 6, the word "or" appears twice)

Response: Thank the reviewer for pointing it out. The intention that "or" appears twice is “and/or” after Eq 6 (now corrected). We have also carefully proofread the paper in this revision.

(c) The authors need to improve the visualizations (e.g., the spots in Fig 2d and 2f should be as easy to discern as the spots in Fig 2c. In Fig 2b, why are there some white marks in the RCTD results?)

Response: Thank you for your constructive suggestions on our visualization. We assume you referred to the low quality of 3d and 3f as the scatter points overlapped, and we have shrunk the point size (also in Extended Fig. 6-7) in this revision. For Fig 2b and 3c, we have also changed them to another dot style for better readability.

Reviewer #2 (Remarks to the Author):

We appreciate the effort that the authors put into revising the manuscript and adapting the tool to best coding practices. However, we still have a few remaining comments.

Major comments:

- We noticed that while the global scores are bound between -1 to +1, the local spatialDM scores range between $-\infty$ to $+\infty$, is this something that is intended and anticipated? We also point the authors to the point of the other reviewer regarding the issue with the normalization term from the local univariate equation. This should not be confused with a priori normalization of the expression.

Response: We much appreciated the reviewer for clarifying the problems with our normalization. In the previous revision, we have implemented a gene-wise standardization for both ligand and receptor expression matrix (highlighted on page 16, blue). This standardization indeed does not guarantee the local R values to be bound to a range of $[-1, 1]$, but it partially addressed the scale issues given that most ($>99.9\%$) Moran's local R values ($R_{i, \text{sender}}$ and $R_{i, \text{receiver}}$) have been refined to the range $(-10, 10)$. To avoid unnecessary extreme outliers, we have further clipped the local R values between -10 and 10 in this revision and the local R values have empirical distribution as shown in Fig. R2 below (for the 8 intestinal samples).

Therefore, the standardization of local R values makes all ligand-receptor pairs comparable, both within and across samples. However, it has a minor sacrifice by losing the information the local communication density of each pair (namely some pairs may have higher expression than others). We think this is an inevitable trade-off for now, hence support both options in the package with the standardization (default) or without it. We have also added this in the Discussion part (p.14).

- Given the large-scale and still ongoing refactoring of the code, have the authors made sure that their initial results are reproducible and the code works? A github link to the analyses

presented with versioning is necessary to ensure best reproducibility practices. Importantly, currently SpatialDM and its tutorial do not work.

Response: Thank the reviewer for pointing it out. We apologized that some codes (e.g., Quick Example) were not updated upon the last submission. We have fixed and tested them throughout and ensure that the package and the tutorial should work now.

Minor comments:

- In their last response the authors mention that “spatial proteomic or with greater resolution” would improve SpatialDM. We are not sure how proteomics is related to the inherent limitations to SpatialDM specifically, rather the authors should address that the normalization of the local variables is based solely according to the mean of each variable, thus it does not e.g. necessarily encode the variation within each variable, etc. In other words, authors should discuss limitations specific to their method, not the field.

Response: Thanks for the suggestions! We would like to clarify that spatial proteomic or with greater resolution will not directly address the inherent limitations of SpatialDM, but only facilitate the interpretations (also highlighted on p. 14). Although SpatialDM has no assumptions on the resolution of each spot, a greater resolution would allow better interpretations given that we also integrated the information of different interaction modes (i.e., long-distance or secreted signalling, and short-distance including cell-cell contact, or ECM). The interactions we attempted to investigate with SpatialDM take place through proteins. SpatialDM does assume a positive correlation between mRNA level and functional protein level, while it is not always the fact given varying translational mechanisms and rates, post-translational modifications, etc.

On the other hand, we agreed with the reviewer that we should further discuss the technical limitation of our method. Here, we have restructured the discussion section by highlighting the local interacting spots detection and expanded our discussions on the effects of its normalization/standardization at a ligand-receptor pair level for the potential impact on local interaction density and p values.

Reviewer #1 (Remarks to the Author):

The authors have satisfactorily addressed our comments.

Reviewer #2 (Remarks to the Author):

We thank the authors for thoroughly addressing our comments. Only one point, yet important, remain: please add a link to a github repository with your analyses to guarantee reproducibility and transparency.

Reviewer #2 (Remarks to the Author):

We thank the authors for thoroughly addressing our comments. Only one point, yet important, remain: please add a link to a github repository with your analyses to guarantee reproducibility and transparency.

Response: Thanks the reviewer for the kind reminder. We have included a link to notebooks for reproducing results in this paper under Code Availability Section (highlighted).